# Piezoelectric MEMS Resonators for Cigarette Particle Detection

**DOI:** 10.3390/mi10020145

**Published:** 2019-02-21

**Authors:** Javier Toledo, Víctor Ruiz-Díez, Maik Bertke, Hutomo Suryo Wasisto, Erwin Peiner, José Luis Sánchez-Rojas

**Affiliations:** 1Microsystems, Actuators and Sensors Group, Universidad de Castilla-La Mancha, 13071 Ciudad Real, Spain; victor.ruiz@uclm.es (V.R.-D.); joseluis.saldavero@uclm.es (J.L.S.-R.); 2Institute of Semiconductor Technology (IHT), Technische Universität Braunschweig, Hans-Sommer-Straße 66, 38106 Braunschweig, Germany and Laboratory for Emerging Nanometrology (LENA), Langer Kamp 6a, 38106 Braunschweig, Germany; m.bertke@tu-braunschweig.de (M.B.); h.wasisto@tu-braunschweig.de (H.S.W.); e.peiner@tu-braunschweig.de (E.P.)

**Keywords:** microelectromechanical systems (MEMS), piezoelectric, aluminum nitride (AlN), low-cost circuit, phase-locked loop, particle

## Abstract

In this work, we demonstrate the potential of a piezoelectric resonator for developing a low-cost sensor system to detect microscopic particles in real-time, which can be present in a wide variety of environments and workplaces. The sensor working principle is based on the resonance frequency shift caused by particles collected on the resonator surface. To test the sensor sensitivity obtained from mass-loading effects, an Aluminum Nitride-based piezoelectric resonator was exposed to cigarette particles in a sealed chamber. In order to determine the resonance parameters of interest, an interface circuit was implemented and included within both open-loop and closed-loop schemes for comparison. The system was capable of tracking the resonance frequency with a mass sensitivity of 8.8 Hz/ng. Although the tests shown here were proven by collecting particles from a cigarette, the results obtained in this application may have interest and can be extended towards other applications, such as monitoring of nanoparticles in a workplace environment.

## 1. Introduction 

Microscopic particles are present in many environments, e.g., the aerospace, healthcare or pharmaceutical industries, and workplaces. For this reason, many studies have been reported about the adverse effects to human health due to their small size and the potential to cause respiratory diseases [1,2,3]. Therefore, there is an increasing need to detect and control the concentration of these particles in multiple contexts, motivating the development of a low-cost system capable of detecting these particles in real-time. Several research groups established various types of micro-/nanoelectromechanical systems (M/NEMS) as mass-sensitive sensors for this purpose [4,5,6,7]. The detection mechanisms and elements of the sensing systems have been varied, e.g., direct optical investigation inside a scanning electron microscope for vertical nanowire resonators [8], square full piezoresistive Wheatstone bridge for out-of-plane cantilevers [9], U-shape piezoresistive Wheatstone bridge for in-plane electrothermal cantilevers [10], and thin piezoelectric films for bulk acoustic devices [11,12]. 

Sampling of particulate matter for gravimetric detection usually requires a particle collector stage, e.g., an impactor [13,14], a thermophoretic [11] or electrostatic sampler [4,8,9], an adhesive layer [15], etc. Basically, that also applies to cigarette smoke particles, which were collected on a silicon cantilever resonator using thermophoresis [11], an impactor [13,14], or an electrostatic field [16]. However, smoke particulates could be sampled without additional measures for particle collection, which was not observed so far for other aerosols.

In this work, the sensor working principle is based on the resonance frequency shift induced by collected particles on a piezoelectric resonator surface [4,16]. For evaluating the sensor sensitivity, the resonator device was exposed to cigarette particles in a sealed chamber, in which the mass-loading effect could subsequently be proven and analyzed. 

## 2. Materials and Setup

For this application, an aluminum nitride (AlN) based piezoelectric resonator was designed, fabricated and optimized for a high-order bending mode. It is well known that increasing mode order, the increase in *Q-factor* and resonance frequency as well as the decrease in effective mass improve the mass sensitivity of the resonator [17,18]. On the other hand, lower frequencies allow easier and cheaper electronic designs for the signal conditioning. In our case, the instrumentation amplifier shows a flat response in the gain curve up to approximately 10 MHz with a phase shift appearing above 1 MHz that affects the resolution of the system. Taking this limitation into account, we chose the fourth order roof-tile shaped vibration mode near 1 MHz. This features five nodal lines in one direction and zero nodal lines in the perpendicular direction. Considering Leissa´s nomenclature [19], the vibration mode is named as 05-mode [20].

For an efficient excitation of higher order vibration modes, an appropriate electrode layout is required so that the target mode is optimally excited and sensed. In particular, roof tile-shaped modes are optimally actuated using a stripped electrode design [12,21], which allows a two-port scheme. In order to prevent excessive damping associated with the substrate, the anchoring of the plates to the substrate frame was carefully designed to reduce the energy dissipation. Beam anchors were placed at the same position as the nodes for a 05-mode, and their resonance frequency was also coincident with that of the target mode. 

One more aspect to consider in the device design is the magnitude of the area exposed to the particles. For this particular vibration mode, the resonance frequency is determined by the width of the anchored side. The length along the nodal lines is chosen to have both good signal to noise ratio [22] and a sensitive area much larger than the particles to be detected, to avoid edge effects. The fabricated resonator and its considered modal shape measured by a laser Doppler vibrometer are shown in Figure 1a,b, respectively. The resonator has a length of *l* = 3000 μm, a width of *w* = 900 μm and a thickness of *t* = 10 μm.

The top metallization has four striped electrodes that allow a selective excitation of the vibration modes and act as a filter for higher modes [23]. The resonator is excited by an anti-parallel connection of the electrodes (+−+−) to obtain the desired modal vibrations. In this case, two top electrodes connected in parallel (+) were used as actuation ports and the other two (−) as sensing ports (see Figure 1b). The structure is anchored with five supports in the nodal lines to resemble a free plate condition. Subsequently, this leads to an increased piezoelectric response compared to a single side clamped cantilever structure with the same geometry and quality factor [24]. The fabrication of the micro-plate resonator was carried out by MEMSCAP (Durham, NC, USA), following the PiezoMUMPs process [25]: a silicon on insulator (SOI) wafer with 10 μm-thick device layer was covered with a 500 nm-thick aluminum nitride (AlN) piezoelectric film. The silicon was doped to serve as both bottom electrode and structural layer. As a top electrode, a layer of aluminum (1 μm) was deposited at the final device processing step. 

The values of the resonant parameters (i.e., resonance frequency (*f*_r_) and quality factor (*Q-factor*)), were registered in air before the deposition of cigarette particles, obtaining values of 1108 kHz and 185, respectively. For this work, only one device was used, but other devices from the same production batch were characterized in terms of *f*_r_ and *Q-factor*. Structures with the same nominal dimensions showed values of 1108 ± 2 kHz and 189 ± 10 for *f*_r_ and *Q-factor*, respectively.

In this work, no system has been implemented regarding the sampling of cigarette particles on the resonator surface [5]. The lack of additional sampling requirements is an advantage of our approach. However, after conducting several experiments, it was observed that despite removing the cigarette particles with N_2_, the initial value of the resonance frequency was shifted. This occurs because the ash particles coming from the cigarette can be easily cleaned with air or N_2_. However, other types of particles exist such as nicotine or tar (tobacco residue) that may adhere to the resonator surface. To avoid this problem, a cleaning process with isopropanol was implemented after different experiments. In addition, the resonator chip could be cleaned with isopropanol without needing to be removed before from its package. This is a clear advantage over approaches where the resonant device is considered to be disposable and, thus, has to be replaced before a decrease of particle capture efficiency is detected [15].

The resonant parameters change due to external environmental influences on the device and need to be determined in real-time. The detection of these parameters can be performed using several methods [4,26,27]. The real-time monitoring of the resonant parameters may be challenging due to the low-quality factors and parasitic effects present in the resonator [12,28]. For this reason, an interface circuit based on resonators in two-port configuration involving a parasitic compensation device and an instrumentation amplifier was used to cancel the parasitic effects (see Figure 2). However, our results show that a capacitive crosstalk across the actuation and sensing ports has a significant contribution to the output current. To minimize this parasitic component, a compensation (dummy) device was introduced [23]. This additional device duplicates the structure of the resonator, being basically an exact copy of the plate resonator, unreleased to prevent its vibration. As it can be observed in Figure 3, the application of the interface circuit results in a clear resonance, with low baseline and the necessary phase to meet the Barkhausen criterion for oscillation. 

In addition, a closed-loop circuit, based on the phase-locked loop (PLL) integrated circuit 74HCT4046 [29], is implemented for tracking the resonant parameters of the resonator. This oscillator circuit is a feedback system composed of a phase comparator (PC2), a low-pass filter (Filter 1) and a voltage-controlled oscillator (VCO). Basically, this circuit is a feedback loop where the VCO can be automatically synchronized with the signal coming from the resonance of the piezoelectric resonator [30,31]. Besides, as it can be seen in Figure 4, the output signal of the VCO (*f*_osc_) was adapted with a low-pass filter (filter 2) since the VCO generates a digital signal. In this way, the excitation of the resonator is carried out with a filtered sinusoidal signal obtaining a higher stability of the oscillation frequency. This oscillator circuit allows us to obtain a reference signal (*f*_ref_), which controls the frequency and phase of the oscillator. In this circuit, the feedback error signal is a phase lag, instead of a voltage or current as in conventional feedback systems. The self-correcting capability of this system allows the PLL to modify the oscillation frequency as a function of mass changes on the resonator surface due to the addition of particles. 

## 3. Measurements and methods

To demonstrate the functionality of the system, the resonator is characterized with different open-loop and closed-loop configurations and experimental arrangements. Finally, the added mass of the cigarette particles on the resonator surface is estimated. 

### 3.1. Open-Loop Measurements

Initially, the 05-mode was analyzed with an open-loop technique based on a USB data acquisition board, *Digilent* instrument *Analog Discovery 2* (Digilent Inc., Pullman, WA, USA) [32], capable of performing sweep frequency measurements. This instrument is connected to a PC in order to calculate *f*_r_ and *Q-factor* of the resonator since these parameters have a large influence on the sensitivity and efficiency of the mass concentration sensing principle. A schematic of the setup along with the interface circuit is presented in Figure 5. The sensor is separated by a distance of 10 cm from the cigarette. This prevents larger particles from depositing directly on the resonator surface. In addition, the sensor is located on the same printed circuit board (PCB) next to the interface circuit.

In the first step, the performance of the resonator was evaluated by collecting cigarette particles on the surface resonator for 15 minutes and subsequently measuring the frequency shift produced. As it can be observed in Figure 3, a frequency shift of around 10 kHz was detected after sampling the particles. Nevertheless, a phase shift of around −15° is observed due to differences between the dummy and resonator, and the bandwidth limitation of the instrumentation amplifier. However, this resonance curve is adequate for the later implementation of the oscillator circuit. 

Once the behaviors of the resonator and electronic system were tested, two similar experiments for the detection of cigarette particles were performed. The main objective was to check the deposition rate of the cigarette particles and the reproducibility of the measurements. In this case, we carried out a frequency sweep every minute. In order to calculate the resonant parameters, the measured impedance spectrum was fitted to a modified Butterworth-Van-Dyke (BVD) equivalent circuit [33]. The frequency resolution measured of the resonator was 70 Hz, a rather high value compared with other results [12,20], associated with the open-loop configuration and the commercial instrument employed in the measurements. 

In Figure 6 and Figure 7, the open-loop measurements are displayed showing the evolution of the resonant parameters with the deposition of cigarette particles. As expected from the previous results, *f*_r_ and *Q-factor* decreased over time due to a higher concentration of cigarette particles on the surface resonator. More deposited cigarette particles result in higher frequency shifts. As it can be observed in Figure 6 and Figure 7, an almost linear decrease occurs for *f*_r_ and *Q-factor* obtaining a decrease of around 625 ± 8 Hz/min and −0.55 ± 0.03 min^−1^, respectively.

The slight linear decrease of *Q*(*t*) observed in Figure 6 and Figure 7 indicates that the roof-tile-shaped vibration mode is sensitive to a damping effect by the deposited cigarette particles. This effect can be explained through Equation (1), which shows that an increase in *Q-factor* might be expected with the added mass of the cigarette particles (*Δm*_cig_) assuming a uniform deposition. [34]. However, their effect in *Q-factor* may also lead to an increased viscous damping ($c_{\mathrm{damp}}$), due to the presence of big particles with irregular shape [9] (see Figure 12).
(1)$$Q\text{-}factor=2\pi f_{\mathrm{can}}m_{\mathrm{can}}\frac{\sqrt{1+\frac{\Delta m_{\mathrm{cig}}}{m_{\mathrm{can}}}}}{c_{\mathrm{damp}}}$$
where *f*_can_, *m*_can_ and *c*_damp_ are the resonator frequency before particle deposition, mass resonator and viscous damping, respectively.

Regarding the evolution of resonance frequency with time, assuming both constant concentration of cigarette particles in the ambient air and constant adsorption rate of these particles on the resonator, we can expect a constant decrease rate Δ*f*_r_/Δ*t* and thus a linear decrease of *f*_r_(*t*) as confirmed by our experiments in Figure 6 and Figure 7. The added mass will be estimated in Section 3.3.

### 3.2. Closed-Loop Measurements

In the previous section, we could check that the open-loop technique presents a poor frequency resolution. In addition, this technique requires a long measurement time, limiting the potential application in a real-time monitoring system. For this reason, a low-cost PLL-based oscillator circuit, utilizing the previously described 74HCT4046, was developed. 

The implementation of an oscillator is challenging due to the hydrodynamic loading (damping in air) and parasitic effects of the resonator, which increase for larger surfaces. For this reason, the previously described interface circuit with dummy compensation was also included in the system. In this setup, the PLL circuit was combined with the resonator and the previous interface circuit as a closed-loop oscillator system as shown in Figure 8. 

In order to obtain the resonant parameters, it was necessary to perform a calibration process [35]. In this process, the oscillator circuit outputs, oscillation frequency (*f*_osc_), and the circuit output gain (*G*_osc_) were transformed into the mechanical magnitudes of the resonator, *f*_r_ and *Q-factor*, using a resonator model [36]. In this case, the commercial instrument was able to obtain the circuit outputs, *f*_osc_ and *G*_osc_, with a rate of 5 samples/s obtaining a better frequency resolution of 1 Hz. To test the system performance, two different experiments were carried out with the closed-loop setup. In the first experiment, particles were collected as in the previous section obtaining a decrease rate for *f*_r_ of around 621 Hz/min (see Figure 9). 

In the second experiment, the sensitivity and response of the system were further checked. In order to do that, we introduced an additional cigarette in the chamber and opened and closed it several times, to assess the response of the system. As it can be observed in Figure 10, several peaks in the resonance curves were detected as a consequence of this. Besides, we could check that the deposition of particles was higher, with a decrease rate for *f*_r_ of around 1690 Hz/min, when two cigarettes were introduced at the same time in the sealed chamber.

The open-loop measurements displayed in Figure 6 and Figure 7 as well as the closed-loop measurements in Figure 9, which were carried out with the smoke of one cigarette in the chamber, show almost monotonic decrease of *f*_r_ and *Q-factor* with time. Fluctuations of both *f*_r_(*t*) and *Q*(*t*) about the fitted straight lines are visible in Figure 9, and Figure 10 shows even slight increases of both parameters caused by opening of the chamber and the corresponding concentration decrease of cigarette particles inside the chamber. We attribute this effect to a transient loss of deposited cigarette smoke from the resonator by evaporation, which is reasonable owing to the considerable amount semivolatile components in cigarette smoke, e.g., nicotine and water [37].

Figure 11 and Figure 12 show sensor top-view images obtained with a scanning electron microscope (SEM). The SEM images depict the resonator covered by particles of different sizes and shapes. The previous measurements show that these particles can be detected in real-time with a low-cost oscillator circuit.

### 3.3. Estimation of Added Mass

Once the performance of the resonator was tested with different open-loop and closed-loop techniques, the mass of the deposited cigarette particles, *Δm*_cig_, can be estimated through Equation (2) [16,38]:(2)$$\Delta m_{\mathrm{cig}}≃\frac{2m_{\mathrm{can}}}{f_{\mathrm{can}}}\Delta f_{\mathrm{cig}}$$
where *Δf*_cig_, *f*_can_, *m*_can_ are the frequency shift after particle deposition, resonator frequency before particle deposition, and mass, respectively.

This expression is under the assumption that the spring constant of the resonator is maintained and that the deposition of particles is uniform on the surfaces. The resonator used in this work has a mass of *m*_can_ = 62.91 µg and a resonance frequency in air of *f*_can_ = 1108 kHz. The added mass was estimated for the second open loop measurement presented in Figure 7. In that case, a frequency shift (*Δf*_cig_) of around 10 kHz was detected after collecting particles for 15 minutes, obtaining a value of *Δm*_cig_ = 1.14 µg. Moreover, the mass-loading sensitivity of the sensor (*Δf*_cig_*/Δm*_cig_) was calculated obtaining a value of 8.8 Hz/ng. This sensitivity value is similar to that reported by Wasisto et al. of 0.15 Hz/ng for a cantilever sensor of smaller dimensions with a lower resonance frequency [38].

This sensitivity was obtained using a USB data acquisition board (Digilent instrument, Digilent Inc., Pullman, WA, USA) and the closed-loop technique with a frequency resolution of 1 Hz. A frequency resolution of around 8 mHz could be achieved with the same device but using a higher-cost instrument such as a lock-in amplifier [39]. With these two values for the resolution in the tracking of resonance frequency and the sensitivity given above, the limit of detection can be 114 pg or 1 pg for the two measurement configurations, respectively. The sensitivity of the resonator was also estimated and compared taking into consideration the surface of the resonator, obtaining a sensitivity ratio of 21.4 μm^2^/ng. This value is also similar to that reported by Wasisto et al. of 8.54 μm^2^/ng [4].

### 3.4. Discussion

M/NEMS-based gravimetric sensors for cigarette smoke detection have been already reported, with different strength and weaknesses. As it can be seen in Table 1, mass sensitivity up to several tens of Hz/pg and resolution of 29.8 fg have been stated, which are clearly better than the values of 8.8 Hz/ng and 114 pg, respectively, obtained in this work. However, our device has a larger surface compared to the size of the particles. In this way, edge effects can be neglected since the particles are uniformly distributed on the resonator surface. 

Besides, the present approach has the advantage that no particle sampler is required due to the large surface of the sensor and frequency tracking is performed using an integrated, low-cost PLL circuit. Furthermore, in our case we can clean the resonator chip without needing to remove it from its package using N_2_ and isopropanol. These issues, although not stressed in the other studies, are directly related to the planned use of such sensors in portable applications, i.e. monitoring of personal aerosol exposure.

## 4. Conclusions

In conclusion, the development of a closed-loop circuit setup based on a piezoelectric micro-plate resonator and the integrated circuit 74HCT4046 has been demonstrated. The system is capable of tracking the resonance frequency and quality factor of the device, oscillating in a high order vibrational mode (05-mode), with a sensitivity of 8.8 Hz/ng and a resolution of 114 pg. In this case, the mass-loading effect tests were proven by collecting particles from a cigarette. Nevertheless, the results obtained in this application may have potential applications for the detection of nanoparticles in a workplace environment.

## Figures and Tables

**Figure 1 micromachines-10-00145-f001:**
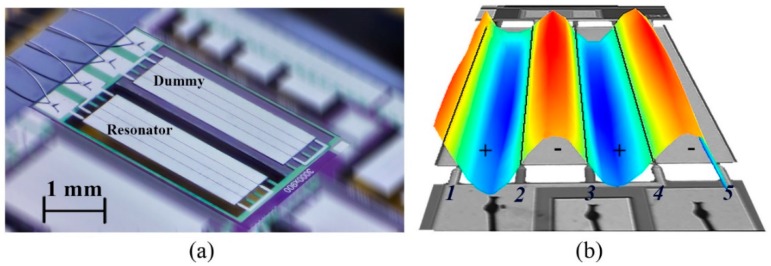
(**a**) Top-view micrograph of the resonator (3000 μm × 900 μm × 10 μm) and compensation device in a dual in-line package. (**b**) Modal shape measured with a laser Doppler vibrometer (05-mode).

**Figure 2 micromachines-10-00145-f002:**
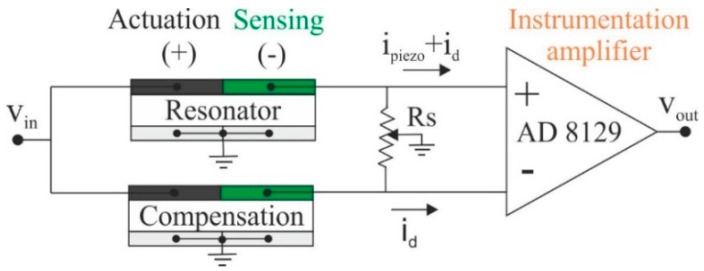
Schematic of interface circuit to measure the 05-mode response by cancelling the parasitic capacitance.

**Figure 3 micromachines-10-00145-f003:**
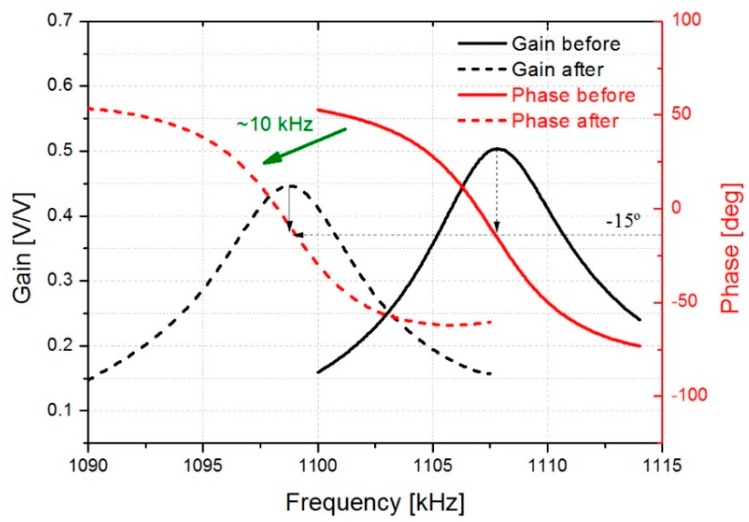
Open loop response of the interface circuit for the 05-mode. Gain-phase before represents the initial state of the resonator in air, and gain-phase after indicates the final state after the deposition of cigarette particles.

**Figure 4 micromachines-10-00145-f004:**
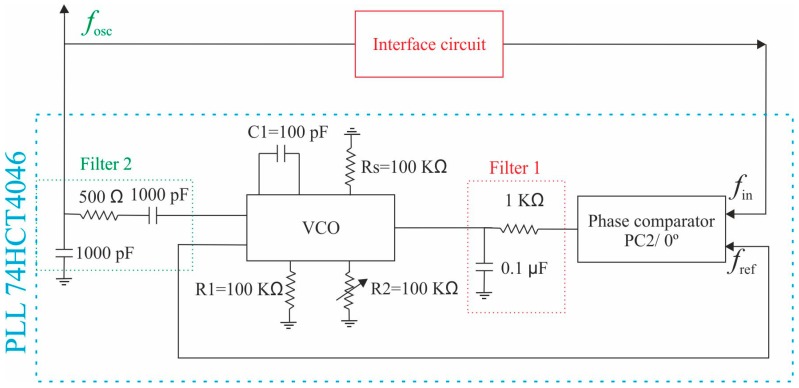
Schematic of phase-locked loop (PLL) 74HCT4046 oscillator circuit including the interface circuit.

**Figure 5 micromachines-10-00145-f005:**
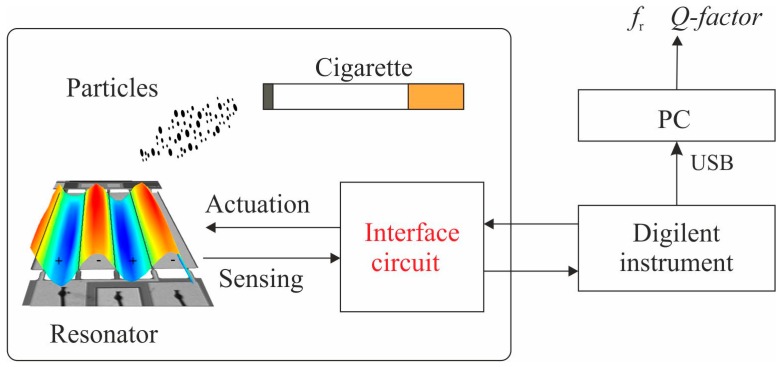
Schematic of the open-loop setup with the Digilent instrument.

**Figure 6 micromachines-10-00145-f006:**
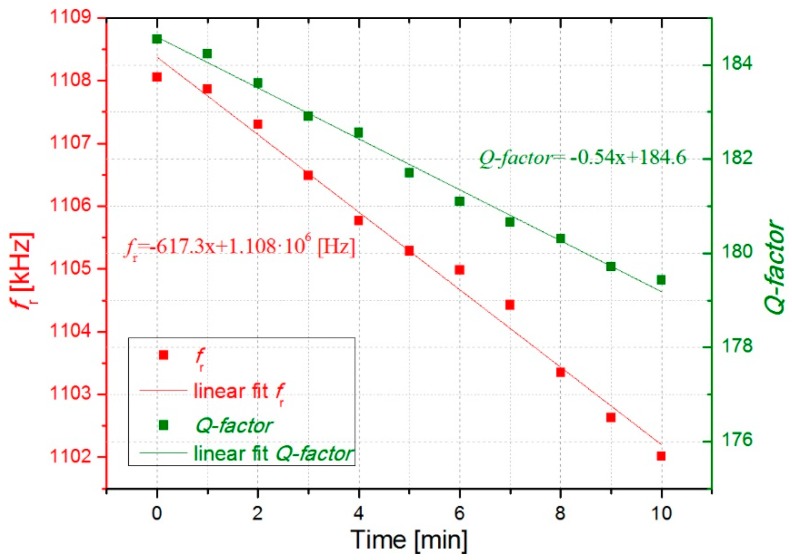
Open loop measurement showing the evolution of the resonant parameters with the deposition of cigarette particles for 10 minutes.

**Figure 7 micromachines-10-00145-f007:**
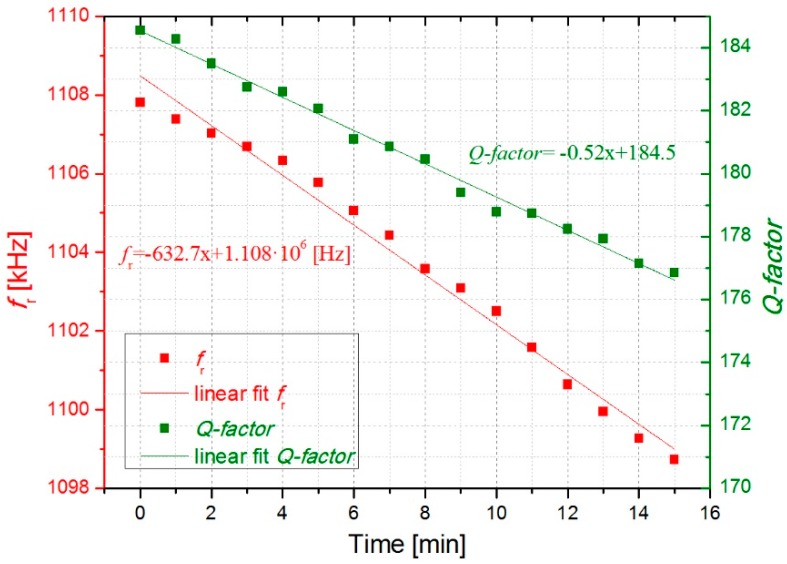
Open loop measurement showing the evolution of the resonant parameters with the deposition of cigarette particles for 15 minutes.

**Figure 8 micromachines-10-00145-f008:**
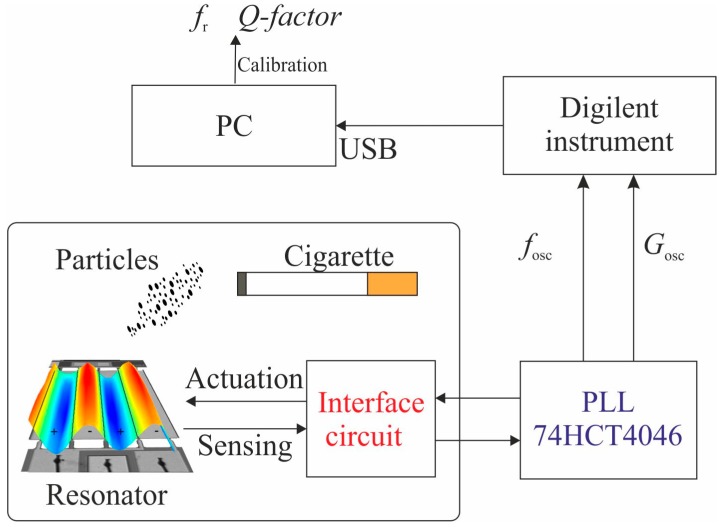
Schematic of the closed-loop setup with the PLL 74HCT4046.

**Figure 9 micromachines-10-00145-f009:**
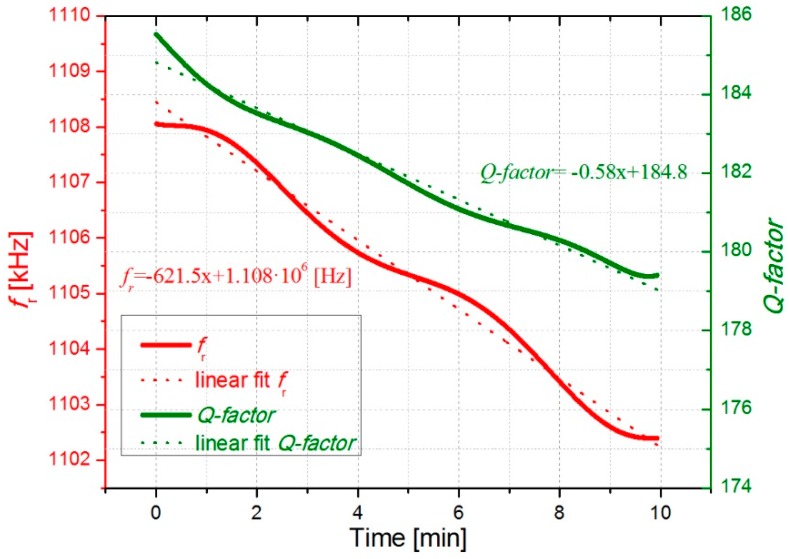
Closed loop measurement showing the evolution of the resonant parameters with the deposition of particles from one cigarette.

**Figure 10 micromachines-10-00145-f010:**
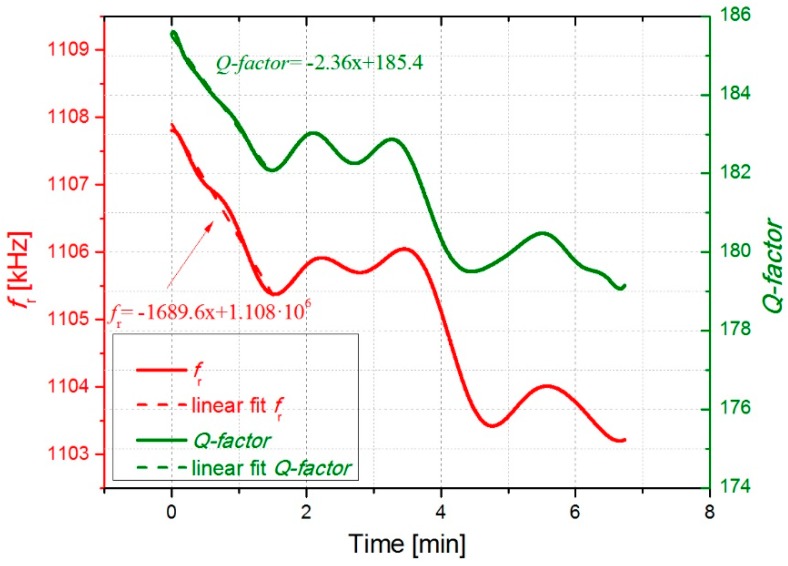
Closed loop measurement showing the evolution of the resonant parameters with the deposition of particles from two cigarettes.

**Figure 11 micromachines-10-00145-f011:**
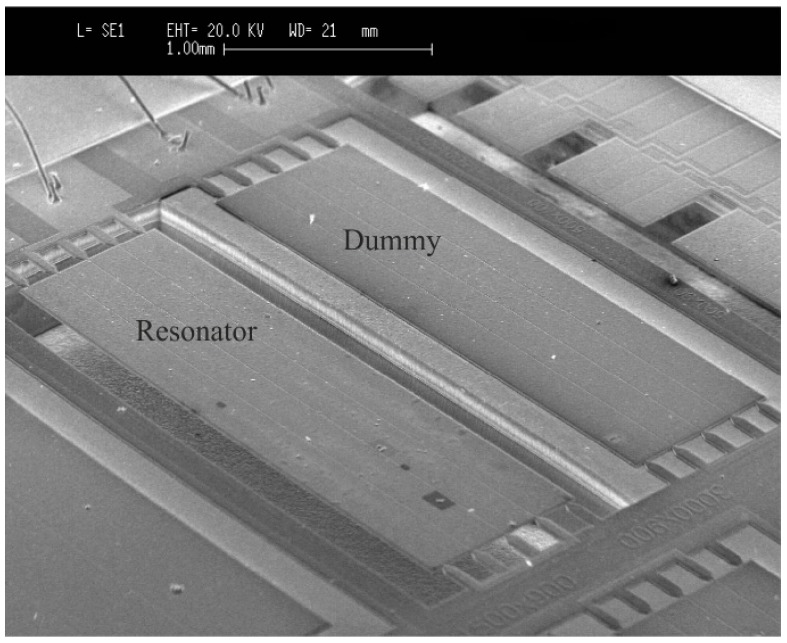
Micrograph of the 05-mode resonator and the dummy device with scanning electron microscope (SEM) after the deposition of cigarette particles.

**Figure 12 micromachines-10-00145-f012:**
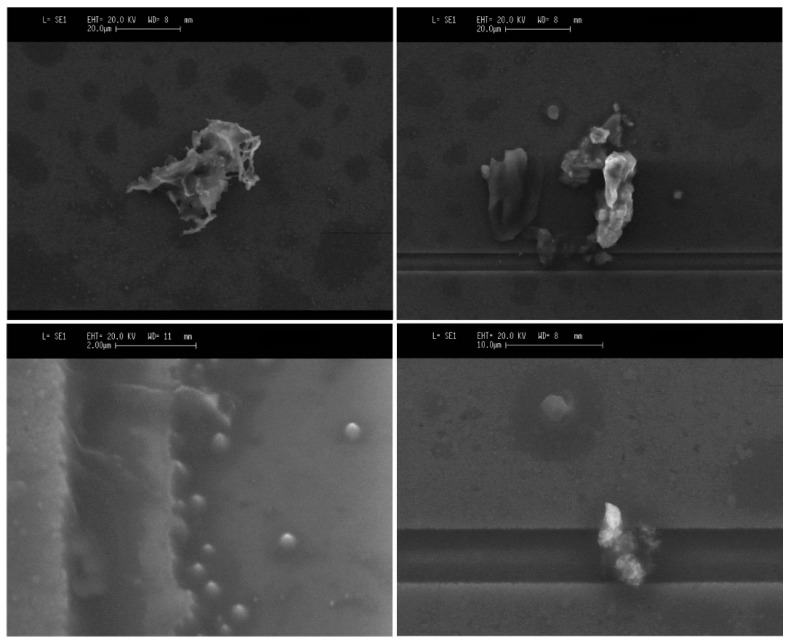
Micrographs of different cigarette particles deposited on the surfaces of a piezoelectric resonator.

**Table 1 micromachines-10-00145-t001:** M/NEMS-based resonators for gravimetric cigarette smoke detection.

M/NEMS Resonator	Particle Sampler	PLL for Frequency Tracking	Mass Sensitivity	Mass Resolution	Reference
FBAR ^1^	thermophoretic	integrated	not given	2 µg/m^3^	[11]
TPR ^2^	impactor	not given	42 Hz/pg	not given	[13]
TPR ^2^	impactor	external	1.9 Hz/pg	29.8 fg	[14]
PCR-NWA ^3^	no	no	20 Hz/ng	50–100 pg	[40]
PCR ^4^	electrostatic	integrated	12 Hz/ng	76.2 pg	[16]
PPR ^5^	no	integrated	8.8 Hz/ng	114 pg	This work

^1^ FBAR: film bulk acoustic resonator. ^2^ TPR: thermal-piezoresistive resonator. ^3^ PCR-NWA: piezoresistive cantilever resonator with nanowire array. ^4^ PCR: piezoresistive cantilever resonator. ^5^ PPR: piezoelectric plate resonator.

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
