# Peer review of "Piezoelectric MEMS Resonators for Cigarette Particle Detection"

_micromachines, 2019, doi:10.3390/mi10020145_

Round 1

Reviewer 1 Report

Very interesting topic, and presentation of the full design of a sensor system.

The paper could be further improved by:

- further explaining/justifying the choice of the resonator structure and resonance mode (fourth order roof-tile shaped vibration mode)

- providing more details about the statistical specs measurements for device resonance frequency and Q-factor. 1108 KHz and 185 seem to be for one device. How many devices were tested. What are the standard deviations?

- giving more insight about the potential use of this system in practice, in particular with respect to its selectivity for detecting nanoparticles of different types (e.g. cigarette particles vs dust vs other, etc)

- clarifying the location of the resonating sensor in figure 4. If it is contained within the interface circuit, this should be specified.

- comparing the performance of the sensor system with at least a few works from the state-of-the-art

Author Response

Dear reviewer,

We greatly appreciate your thoughtful comments that helped to improve the manuscript. We trust that all of your comments have been addressed accordingly in a revised manuscript. Thank you very much for your effort.

In the following, we give a point-by-point reply to your comments (bold type), with our answer (blue colour) and the amendments to the manuscript (green colour):

Comments and Suggestions for Authors

Very interesting topic, and presentation of the full design of a sensor system.

The paper could be further improved by:

1. Further explaining/justifying the choice of the resonator structure and resonance mode (fourth order roof-tile shaped vibration mode)

It is well known that the resonator sensitivity increases with the order of the mode. Besides, the electronics introduces some limitations to the increase of the order and frequency of the mode. A low frequency value is necessary since the instrumentation amplifier shows a flat response in the gain curve up to approximately 10 MHz. If this value is exceeded, a phase shift is introduced at the resonance frequency, decreasing the resolution of the system. On the other hand, we have the bandwidth limitation of the Digilent instrument near 30 MHz. Therefore, a resonance frequency around 1 MHz, such as the device used in this work, is suitable for both the electronic circuit and the measuring instrument. This frequency is attained with the fourth order mode. Regarding the structure, the shape of the electrodes and anchoring of the plates have been optimized to reduce the losses and obtain the best possible signal.

In order to justify the selection of the resonator and vibration mode, we have added the following text in section 2.

“It is well known that increasing mode order, the increase in Q-factor and resonance frequency as well as the decrease in effective mass improve the mass sensitivity of the resonator [17,18]. On the other hand, lower frequencies allow easier and cheaper electronic designs for the signal conditioning. In our case, the instrumentation amplifier shows a flat response in the gain curve up to approximately 10 MHz with a phase shift appearing above 1 MHz that affects the resolution of the system. Taking this limitation into account, we chose the fourth order roof-tile shaped vibration mode near 1 MHz. This features five nodal lines in one direction and zero nodal lines in the perpendicular direction. Considering Leissa´s nomenclature [19], the vibration mode is named as 05-mode [20].

 For an efficient excitation of higher order vibration modes, an appropriate electrode layout is required so that the target mode is optimally excited and sensed. In particular, roof tile-shaped modes are optimally actuated using a stripped electrode design [12,21], which allows a two-port scheme. In order to prevent excessive damping associated with the substrate, the anchoring of the plates to the substrate frame was carefully designed to reduce the energy dissipation. Beam anchors were placed at the same position as the nodes for a 05-mode, and their resonance frequency was also coincident with that of the target mode.

One more aspect to consider in the device design is the magnitude of the area exposed to the particles. For this particular vibration mode, the resonance frequency is determined by the width of the anchored side. The length along the nodal lines is chosen to have both good signal to noise ratio [22] and a sensitive area much larger than the particles to be detected, to avoid edge effects.”

2. Providing more details about the statistical specs measurements for device resonance frequency and Q-factor. 1108 KHz and 185 seem to be for one device. How many devices were tested? What are the standard deviations?

In this work, the complete characterization and applications described in the paper were carried out in one single device. However, other resonators from the same production batch were assessed in terms of resonance frequency and quality factor, showing similar values and standard deviations. We have added the following information in section 2.

“For this work, only one device was used, but other devices from the same production batch were characterized in terms of fr and Q-factor. Structures with the same nominal dimensions showed values of 1108 ± 2 kHz and 189 ± 10 for fr and Q-factor, respectively.”

3. Giving more insight about the potential use of this system in practice, in particular with respect to its selectivity for detecting nanoparticles of different types (e.g. cigarette particles vs dust vs other, etc)

The system developed in this work does not allow to discriminate the origin of the particles. Nevertheless, cigarette particles were easily sampled and detected thanks to the large surface and sensitivity of the resonator. To extend the range of applications, a particle attraction or filtering system is required in order to analyze the particles of interest. In order to clarify this question, we have added the following information in section 1.

“Sampling of particulate matter for gravimetric detection usually requires a particle collector stage, e.g., an impactor [13, 14], a thermophoretic [11] or electrostatic sampler [4,8,9], an adhesive layer [15], etc. Basically, that also applies to cigarette smoke particles, which were collected on a silicon cantilever resonator using thermophoresis [11], an impactor [13,14], or an electrostatic field [16]. However, smoke particulates could be sampled without additional measures for particle collection, which was not observed so far for other aerosols.”

4. Clarifying the location of the resonating sensor in figure 4. If it is contained within the interface circuit, this should be specified.

The following information has been added in section 3.1:

“The sensor is separated by a distance of 10 cm from the cigarette. This prevents larger particles from depositing directly on the resonator surface. In addition, the sensor is located on the same printed circuit board (PCB) next to the interface circuit.

5. Comparing the performance of the sensor system with at least a few works from the state-of-the-art

A comparison between different resonators for cigarette particle detection and the proposed one has been added in a new section 3.4 Discussion.

3.4. Discussion

“M/NEMS-based gravimetric sensors for cigarette smoke detection have been already reported, with different strength and weaknesses. As it can be seen in Table 1, mass sensitivity up to several tens of Hz/pg and resolution of 29.8 fg have been stated, which are clearly better than the values of 8.8 Hz/ng and 114 pg, respectively, obtained in this work. However, our device has a larger surface compared to the size of the particles. In this way, edge effects can be neglected since the particles are uniformly distributed on the resonator surface.

Table 1. M/NEMS-based resonators for gravimetric cigarette smoke detection.

M/NEMS   resonator

Particle   sampler

PLL for   frequency tracking

Mass   sensitivity

Mass   resolution

Reference

FBAR1

thermophoretic

integrated

not given

2 µg/m3

[11]

TPR2

impactor

not given

42 Hz/pg

not given

[13]

TPR2

impactor

external

1.9 Hz/pg

29.8 fg

[14]

PCR-NWA3

no

no

20 Hz/ng

50-100 pg

[41]

PCR4

electrostatic

integrated

12 Hz/ng

76.2 pg

[16]

PPR5

no

integrated

8.8 Hz/ng

114 pg

This work

1FBAR: film bulk acoustic resonator. 2TPR: thermal-piezoresistive resonator. 3PCR-NWA: piezoresistive cantilever resonator with nano wire array. 4PCR: piezoresistive cantilever resonator. 5PPR: piezoelectric plate resonator

Besides, the present approach has the advantage that no particle sampler is required due to the large surface of the sensor and frequency tracking is performed using an integrated, low-cost PLL circuit. Furthermore, in our case we can clean the resonator chip without needing to remove it from its package using N2 and isopropanol. These issues, although not stressed in the other studies, are directly related to the planned use of such sensors in portable applications, i.e. monitoring of personal aerosol exposure.”

Reviewer 2 Report

The manuscript demonstrates a resonant sensor which targets the application of sensing cigarette particles. Besides, an interface circuit was implemented to monitor the resonance parameters. This topic is interesting and valuable to M/NEMS community. However, there are some critical issues the authors need to solve before this manuscript becomes acceptable:

1.         There are lots of reported work that use M/NEMS resonators to realize mass sensors. It is highly expected that authors can make a table comparing the performances of the state-of-the-art against the proposed one.

2.         The proposed sensor works based on mass-loading effect, which means deposited particles other than cigarette should also result in frequency shift. How to distinguish cigarette particles from other particles? Is there any special design in the proposed device regarding this problem? If not, it is problematic to say this type of device is specially designed for cigarette particles.

3.         There are two important parameters for sensors, namely sensitivity and limit of detection. Limit of detection is related to noise of the device and electronic circuits. It would be great if authors can also analyze the noise of the system and provide the figure for limit of detection.

4.         Page 2 line 47: it is too vague to conclude that “lower frequencies allow easier and cheaper electronic designs ……”. Authors are expected to provide a more specific frequency range.

5.         It is expected that authors could provide a motivation of choosing roof-tile shaped vibration mode. What is the benefit of this vibration mode compared to other modes such as Lame wave mode which is typically used for plate structure. Is the chosen mode more sensitive?

6.         In figure 1(b), two outmost nodal lines are not correctly positioned. They should in line with anchors. It would be better to label the nodal lines.

7.         Page 2 line 74, it is stated that “……removing the cigarette particles with N2, the initial value of the resonance frequency was shifted. To avoid this problem, a cleaning process with isopropanol was implemented after the different experiments.” Is it possible for authors to explain why this happens? Is it possible to use isopropanol liquid to clean the surface after devices are released? It would be helpful that authors give more information on this regard.

8.         In figure 3, it is very confusing to use “Gain before” and “Gain after” in legend. If there is limited space, authors can give detail of the curves in the caption of the figure.

9.         In figure 6 and 7, it is expected that author can give some explanations on why the resonant frequency and Q decrease linearly with the amount of deposited mass. It would be better to provide some equations in this regard.

10.     In figure 9 and 10, the measured curves are no longer monotonic. Authors are expected to provide detailed explanation on this observation, because it is the monotony that promises the one-to-one relationship between resonant frequency and the deposited mass. This problem is vital to whole work.

Author Response

Dear reviewer,

We greatly appreciate your thoughtful comments that helped improve the manuscript. We trust that all of your comments have been addressed accordingly in a revised manuscript. Thank you very much for your effort.

In the following, we give a point-by-point reply to your comments (bold type), with our answer (blue colour) and the amendments to the manuscript (green colour):

Comments and Suggestions for Authors

The manuscript demonstrates a resonant sensor which targets the application of sensing cigarette particles. Besides, an interface circuit was implemented to monitor the resonance parameters. This topic is interesting and valuable to M/NEMS community. However, there are some critical issues the authors need to solve before this manuscript becomes acceptable:

1. There are lots of reported work that use M/NEMS resonators to realize mass sensors. It is highly expected that authors can make a table comparing the performances of the state-of-the-art against the proposed one.

A summary Table 1 and the following information have been added in a new section 3.4 Discussion, making a comparison between different resonators for cigarette particle detection and the proposed one.

3.4. Discussion

“M/NEMS-based gravimetric sensors for cigarette smoke detection have been already reported, with different strength and weaknesses. As it can be seen in Table 1, mass sensitivity up to several tens of Hz/pg and resolution of 29.8 fg have been stated, which are clearly better than the values of 8.8 Hz/ng and 114 pg, respectively, obtained in this work. However, our device has a larger surface compared to the size of the particles. In this way, edge effects can be neglected since the particles are uniformly distributed on the resonator surface.

Table 1. M/NEMS-based resonators for gravimetric cigarette smoke detection.

M/NEMS   resonator

Particle   sampler

PLL for   frequency tracking

Mass   sensitivity

Mass   resolution

Reference

FBAR1

thermophoretic

integrated

not given

2 µg/m3

[11]

TPR2

impactor

not given

42 Hz/pg

not given

[13]

TPR2

impactor

external

1.9 Hz/pg

29.8 fg

[14]

PCR-NWA3

no

no

20 Hz/ng

50-100 pg

[41]

PCR4

electrostatic

integrated

12 Hz/ng

76.2 pg

[16]

PPR5

no

integrated

8.8 Hz/ng

114 pg

This work

1FBAR: film bulk acoustic resonator. 2TPR: thermal-piezoresistive resonator. 3PCR-NWA: piezoresistive cantilever resonator with nano wire array. 4PCR: piezoresistive cantilever resonator. 5PPR: piezoelectric plate resonator

Besides, the present approach has the advantage that no particle sampler is required due to the large surface of the sensor and frequency tracking is performed using an integrated, low-cost PLL circuit. Furthermore, in our case we can clean the resonator chip without needing to remove it from its package using N2 and isopropanol. These issues, although not stressed in the other studies, are directly related to the planned use of such sensors in portable applications, i.e. monitoring of personal aerosol exposure.”

2. The proposed sensor works based on mass-loading effect, which means deposited particles other than cigarette should also result in frequency shift. How to distinguish cigarette particles from other particles? Is there any special design in the proposed device regarding this problem? If not, it is problematic to say this type of device is specially designed for cigarette particles.

The system developed in this work does not allow to discriminate the origin of the particles. Nevertheless, cigarette particles were easily sampled and detected thanks to the large surface and sensibility of the resonator. To increase the potential use of the application a particle attraction or filtering system is required in order to analyze the particles of interest. In order to clarify this comment, we have added the following information in section 1.

“Sampling of particulate matter for gravimetric detection usually requires a particle collector stage, e.g., an impactor [13, 14], a thermophoretic [11] or electrostatic sampler [4,8,9], an adhesive layer [15], etc. Basically, that also applies to cigarette smoke particles, which were collected on a silicon cantilever resonator using thermophoresis [11], an impactor [13,14], or an electrostatic field [16]. However, smoke particulates could be sampled without additional measures for particle collection, which was not observed so far for other aerosols.”

3. There are two important parameters for sensors, namely sensitivity and limit of detection. Limit of detection is related to noise of the device and electronic circuits. It would be great if authors can also analyze the noise of the system and provide the figure for limit of detection.

As the reviewer points out, the limit of detection is associated with the uncertainty in the measured resonant frequency. In closed loop, with the Digilent instrument, this is about 1 Hz. This can be improved to around 8 mHz in a high viscous liquid but using a benchtop instrument such as a lock-in amplifier. We have added the following information in section 3.3.

“This sensitivity was obtained using a USB data acquisition board (Digilent instrument) and the closed-loop technique with a frequency resolution of 1 Hz. A frequency resolution of around 8 mHz could be achieved with the same device, but using a higher-cost instrument such as a lock-in amplifier [39]. With these two values for the resolution in the tracking of resonance frequency and the sensitivity given above, the limit of detection can be 114 pg or 1 pg for the two measurement configurations, respectively. The sensitivity of the resonator was also estimated and compared taking into consideration the surface of the resonator, obtaining a sensitivity ratio of 214,4 cm2/g. This value is also similar to that reported by Wasisto et al. of 85.4 cm2/g [40].

4. Page 2 line 47: it is too vague to conclude that “lower frequencies allow easier and cheaper electronic designs ……”. Authors are expected to provide a more specific frequency range.

In order to provide a more specific frequency range, we have added the following information in section 2.

“On the other hand, lower frequencies allow easier and cheaper electronic designs for the signal conditioning. In our case, the instrumentation amplifier shows a flat response in the gain curve up to approximately 10 MHz with a phase shift appearing above 1 MHz that affects the resolution of the system. Taking this limitation into account, we chose the fourth order roof-tile shaped vibration mode near 1 MHz. This features five nodal lines in one direction and zero nodal lines in the perpendicular direction. Considering Leissa´s nomenclature [19], the vibration mode is named as 05-mode [20].”

5. It is expected that authors could provide a motivation of choosing roof-tile shaped vibration mode. What is the benefit of this vibration mode compared to other modes such as Lame wave mode which is typically used for plate structure? Is the chosen mode more sensitive?

In order to clarify this comment, we have added the following information in section 2. In addition, we have also estimated and compared the sensitivity of the resonator taking into consideration the surface of the resonator.

“It is well known that increasing mode order, the increase in Q-factor and resonance frequency as well as the decrease in effective mass improve the mass sensitivity of the resonator [17,18].

For an efficient excitation of higher order vibration modes, an appropriate electrode layout is required so that the target mode is optimally excited and sensed. In particular, roof tile-shaped modes are optimally actuated using a stripped electrode design [12,21], which allows a two-port scheme. In order to prevent excessive damping associated with the substrate, the anchoring of the plates to the substrate frame was carefully designed to reduce the energy dissipation. Beam anchors were placed at the same position as the nodes for a 05-mode, and their resonance frequency was also coincident with that of the target mode.

One more aspect to consider in the device design is the magnitude of the area exposed to the particles. For this particular vibration mode, the resonance frequency is determined by the width of the anchored side. The length along the nodal lines is chosen to have both good signal to noise ratio [22] and a sensitive area much larger than the particles to be detected, to avoid edge effects.”

 6. In figure 1(b), two outmost nodal lines are not correctly positioned. They should in line with anchors. It would be better to label the nodal lines.

Figure 1 (b) has been modified indicating the nodal lines:

7. Page 2 line 74, it is stated that “……removing the cigarette particles with N2, the initial value of the resonance frequency was shifted. To avoid this problem, a cleaning process with isopropanol was implemented after the different experiments.” Is it possible for authors to explain why this happens?

In order to clarify this question, we have added the following text in section 2:

“This occurs because the ash particles coming from the cigarette can be easily cleaned with air or N2. However, other types of particles exist such as nicotine or tar (tobacco residue) that may adhere to the resonator surface.”

Is it possible to use isopropanol liquid to clean the surface after devices are released?

It would be helpful that authors give more information on this regard.

“In addition, the resonator chip could be cleaned with isopropanol without needing to be removed before from its package. This is a clear advantage over approaches where the resonant device is considered be disposable and thus has to be replaced before a decrease of particle capture efficiency is detected [15].

8. In figure 3, it is very confusing to use “Gain before” and “Gain after” in legend. If there is limited space, authors can give detail of the curves in the caption of the figure.

The figure caption has been modified as follows:

Figure 3. Open loop response of the interface circuit for the 05-mode. Gain-phase before represents the initial state of the resonator in air, and gain-phase after indicates the final state after the deposition of cigarette particles.

9. In figure 6 and 7, it is expected that author can give some explanations on why the resonant frequency and Q decrease linearly with the amount of deposited mass. It would be better to provide some equations in this regard.

This effect can be explained through equation 1 and 2 presented in section 3.1 and 3.3. An increase of cigarette particles deposited on the resonator surface produce a change of the resonant parameters proportional to the added mass. In order to clarify these figures, we have added the following information in section 3.1.

“In Figures 6 and 7, the open-loop measurements are displayed showing the evolution of the resonant parameters with the deposition of cigarette particles. As expected from the previous results, fr and Q-factor decreased over time due to a higher concentration of cigarette particles on the surface resonator. More deposited cigarette particles result in higher frequency shifts. As it can be observed in Figures 6 and 7, an almost linear decrease occurs for fr and Q-factor obtaining a decrease of around 625 ± 8 Hz/min and -0.55 ± 0.03 min-1, respectively.”

“The slight linear decrease of Q(t) observed in Figures 6 and 7 indicates that the roof-tile-shaped vibration mode is sensitive to a damping effect by the deposited cigarette particles. This effect can be explained through Equation (1), which shows that an increase in Q-factor might be expected with the added mass of the cigarette particles (Δmcig) assuming a uniform deposition. [34]. However, their effect in Q-factor may also lead to an increased viscous damping, due to the presence of big particles with irregular shape [9] (see Figure 12).

(See equation (1) in doc file)

where fcan, mcan and cdamp are the resonator frequency before particle deposition, mass resonator and viscous damping, respectively.

Regarding the evolution of resonance frequency with time, assuming both constant concentration of cigarette particles in the ambient air and constant adsorption rate of these particles on the resonator, we can expect a constant decrease rate Δfrt and thus a linear decrease of fr(t) as confirmed by our experiments in Figures 6 and 7. The added mass will be estimated in section 3.3.”

10. In figure 9 and 10, the measured curves are no longer monotonic. Authors are expected to provide detailed explanation on this observation, because it is the monotony that promises the one-to-one relationship between resonant frequency and the deposited mass. This problem is vital to whole work.

In order to clarify the previous comment, we have added the following explanation in section 3.2.

“The open-loop measurements displayed in Figures 6 and 7 as well as the closed-loop measurements in Figure 9, which were carried out with the smoke of one cigarette in the chamber, show almost monotonic decrease of fr and Q-factor with time. Fluctuations of both fr(t) and Q(t) about the fitted straight lines are visible in Figure 9, and Figure 10 shows even slight increases of both parameters caused by opening of the chamber and the corresponding concentration decrease of cigarette particles inside the chamber. We attribute this effect to a transient loss of deposited cigarette smoke from the resonator by evaporation, which is reasonable owing to the considerable amount semivolatile components in cigarette smoke, e.g., nicotine and water [37].

Round 2

Reviewer 1 Report

Interesting work worth publishing. Congratulations 

Reviewer 2 Report

Authors have made significant improvement and addressed all my concerns.